# Establishment of a Cell Suspension Culture of *Ageratina pichinchensis* (Kunth) for the Improved Production of Anti-Inflammatory Compounds

**DOI:** 10.3390/plants9101398

**Published:** 2020-10-21

**Authors:** Mariana Sánchez-Ramos, Laura Alvarez, Antonio Romero-Estrada, Antonio Bernabé-Antonio, Silvia Marquina-Bahena, Francisco Cruz-Sosa

**Affiliations:** 1Departamento de Biotecnología, Universidad Autónoma Metropolitana-Iztapalapa, San Rafael Atlixco 186, Col. Vicentina 09340 Ciudad de México, Mexico; marianasan_06@hotmail.com; 2Centro de Investigaciones Químicas-IICBA, Universidad Autónoma del Estado de Morelos, Avenida Universidad 1001, Col. Chamilpa 62209 Cuernavaca, Morelos, Mexico; lalvarez@uaem.mx; 3Departamento de Madera, Celulosa y Papel, Centro Universitario de Ciencias Exactas e Ingenierías, Universidad de Guadalajara, Km. 15.5, Carretera Guadalajara-Nogales, Col. Las Agujas 45020 Zapopan, Jalisco, Mexico; are@uaem.mx (A.R.-E.); bernabe_aa@hotmail.com (A.B.-A.)

**Keywords:** cell suspension culture, anti-inflammatory activity, phytochemical analysis

## Abstract

*Ageratina pichinchensis* (Kunth) is a plant used in traditional Mexican medicine to treat multiple ailments. However, there have not been biotechnological studies on producing compounds in in vitro cultures. The aim of this study was to establish a cell suspension culture of *A. pichinchensis*, quantify the anti-inflammatory constituents 2,3-dihydrobenzofuran (2) and 3-epilupeol (3), evaluate the anti-inflammatory potential of its extracts, and perform a phytochemical analysis. Cell suspension cultures were established in a MS culture medium of 30-g L^−1^ sucrose, 1.0-mg L^−1^ α-naphthaleneacetic acid, and 0.1-mg L^−1^ 6-furfurylaminopurine. The ethyl acetate extract of the cell culture analyzed by gas chromatography (GC) revealed that the maximum production of anti-inflammatory compounds 2 and 3 occurs on days eight and 16, respectively, improving the time and previously reported yields in callus cultures. The anti-inflammatory activity of these extracts exhibited a significant inhibition of nitric oxide (NO) production. Furthermore, a phytochemical study of the ethyl acetate (EtOAc) and methanol (MeOH) extracts from day 20 led to the identification of 17 known compounds. The structures of the compounds were assigned by an analysis of 1D and 2D NMR data and the remainder by GC–MS. This is the first report of the production of (-)-Artemesinol, (-)-Artemesinol glucoside, encecalin, and 3,5-diprenyl-acetophenone by a cell suspension culture of *A. pichinchensis*.

## 1. Introduction

The genus *Ageratina* (Asteraceae) consists of about 1200 species and is distributed in temperate and subtropical regions of the America, Europe, Africa, and Asia [1,2,3,4]; in Mexico, about 164 species of *Ageratina* have been reported [5,6]. Several species of this genus have been studied, and these studies have demonstrated bactericidal, antifungal, antiviral, analgesic, cytotoxic, and anti-inflammatory effects, as well as the ability to treat gastric ulcers [7,8,9,10,11,12]. In the state of Morelos, Mexico, *Ageratina pichinchensis* is traditionally used to treat gastric ulcers and heal deep wounds. Phytochemically, the aerial parts of *A. pichinchensis* are characterized by containing sterols, triterpenes, benzochromenes, and benzofurans [13,14]. The benzochromenes isolated from *A. pichinchensis* showed insecticidal [12], antifungal [15,16], and gastroprotective activities [17]. However, the production of these compounds can be unsustainable through the large-scale planting of *A. pichinchensis*, because many secondary metabolites tend to vary their production according to climatic fluctuations, as has happened with other species [18].

Numerous studies have shown that plant cell cultures could play an appropriate role in the constant and controlled production of bioactive compounds [19,20,21,22,23]. For example, the production of the anticancer anthraquinone shikonin is obtained in 1–2% yield from the roots of the Asian plant *Lithospermum erythrorhizon*; however, its production depends on the geographical distribution and climate, and in addition, the wild plant requires five–seven years of growth for its production, while cultivated plant cells can produce 12–20% [24,25,26]. Examples like this support the use of plant cells to produce value-added compounds.

Recently, our research group reported the isolation and anti-inflammatory effect of (*2S*,*3R*)-5-acetyl-7,3*α*-dihydroxy-2*β*-(1-isopropenyl)-2,3-dihydrobenzofuran (2) and 3-epilupeol (3) from the ethyl acetate extract of a callus culture of *A. pichinchensis* [27].

In this study, we established a cell suspension culture of *A. pichinchensis* and improved the production of compounds 2 and 3. Furthermore, the extracts were evaluated for their anti-inflammatory activity. The phytochemical analysis of the ethyl acetate and methanol extracts of this culture allowed for the isolation and identification of 17 known accumulated compounds, including the anti-inflammatory compounds 2 and 3. Some of the identified compounds have been described as having anti-inflammatory and antimicrobial activity.

## 2. Results and Discussion

### 2.1. Cell Suspension Culture and Growth Kinetics

It has been shown that the friable callus is most suitable for the establishment of cell suspension cultures of any plant species. *A. pichinchensis* cells that transferred to Murashige and Skoog (MS) liquid culture medium with the same growth regulators as in callus cultures (1.0-mg L^−1^ α-naphthaleneacetic acid (NAA) and 0.1-mg L^−1^ 6-furfurylaminopurine (KIN)) grew easily, showing abundant biomass and a slightly yellowish appearance (Figure 1a–c).

Similar results were observed in cell suspension cultures of *Stevia rebaudiana* that were disintegrated within seven days, and the cells also acquired a yellowish appearance [28]. In this study, the cell growth of *A. pichinchensis* was faster (22 days) in the MS liquid culture medium than in callus cultures; in contrast, callus reached their maximum growth after 30–40 days [27]. This might be caused by the facilitated absorption of nutrients in the liquid medium [29].

The growth kinetic of *A. pichinchensis* cell suspension cultures was maintained for 22 days, during which it showed a typical growth curve (Figure 2). The growth kinetic was characterized by a lag phase of four days, reaching 2.37-g L^−1^ dry weight (DW); subsequently, the cells entered the exponential phase and lasted until day 16. During this time, a maximum biomass accumulation was observed (13.28-g L^−1^) of about 5.6-fold the initial dry weight. The specific growth rate (µ) was 0.20 days^−1^, and doubling time (Dt) was 3.01 days^−1^ until reaching a stationary phase, in which the cell culture showed a brown appearance and decreased growth (Figure 2).

Regarding the consumption of total sugars, an abrupt decrease in the sugar content was observed until day six, but the biomass increased; the remainder of the sugar was stable after day 14 (Figure 2), perhaps due to the consumption of nutrients and lack of oxygen in the medium [30,31].

The growth kinetics is similar to that for *Spilanthes acmella*, in which the cell suspension culture reached a specific growth rate of 0.28 days^−1^; during the exponential phase, the doubling time was 2.50 days^−1^, and the maximum biomass was 8.5-g L^−1^ at day 15 [32]. On the other hand, *Satureja khuzistanica* cell suspension cultures reached a maximum dry biomass of 19.7-g L^−1^ at 21 days, with a specific growth rate of 1.5 days^−1^ and a doubling time of 7.6 days^−1^ [33]. Similarly, cell suspension cultures of *Helianthus annuus* produced 12.7-g L^−1^ dry biomass after nine days of culture, showing a specific growth rate of 0.21 days^−1^ and a doubling time of 3.31 days^−1^ [34]. These results suggest that *A. pichinchensis* cell cultures have a similar tendency as other cell suspension cultures, and, although the biomass yield differs for each species, it can also be produced in a short time.

### 2.2. Cell Viability and pH in the Culture Medium

Cell viability was suitable in the cell suspension culture, decreasing only slightly during the 22 days of culture. On the first day of culture, 89.57% of the viability was present, and by 22 days, it only decreased to 77.19% (Figure 3).

These results confirm that the establishment of the *A. pichinchensis* cell suspension culture was successful since the viability was greater than 50%. In other species, such as *Taxus globose*, the viability decreases to 45–50% at the end of the growth culture [35], which was lower than that of the *A. pichinchensis* culture. The microscopic image shows that the *A. pichinchensis* cell suspension culture produced single cells and some small aggregates (Figure 1d). In a study conducted in *Taxus cuspidate* cell cultures [35], it was found that the size of cell aggregates is important, since at average sizes greater than 800 µm, there is a decrease in the production of paclitaxel; however, in our study, the aggregates were smaller (Figure 1d).

On the other hand, there was a gradual increase in the pH values (5.0–5.3) during the exponential phase (from day four to 18); later, between days 18 and 22, the pH values remained stable at 5.3, but these changes did not affect the biomass or the production of compounds 2 and 3 during the growth kinetics. Likewise, the cell viability experienced a similar behavior; at the beginning of the kinetics, the cell viability was 89%, remaining stable until day 10 (exponential phase), and on subsequent days, a gradual decrease was observed up to 77%, possibly as a consequence of the depletion of nutrients and alkalinization of the culture medium (Figure 3).

In cell suspension cultures of *Dioscorea deltoidea*, there was an acidification of the culture medium during the lag phase, an alkalization during the exponential phase, and a steady state of pH in the stationary phase. At the same time, the exponential growth of the cell culture medium may be associated with the development of alkaline reactions that occur during metabolism [36]. The observed changes in pH values may be related to the use of ammonium and nitrate or the absorption of sugar by the transport mechanism with H^+^ [37]. Moreover, plant cells can also modify the external pH; they can increase or decrease the values according to the pH range in which they are grown, until a balance is produced [38].

### 2.3. Phytochemical Analysis of the Cell Suspension Cultures

The phytochemical analysis of the ethyl acetate (EtOAc) and methanol (MeOH) extracts obtained from the biomass on day 20 led to the identification of 17 known compounds (Figure 4), including the benzochromenes: desmethoxyencecalin (1), (-)-Artemesinol (6), (-)-Artemesinol glucoside (7), and encecalin (10); three triterpenes: 3-epilupeol (3), 24-metilene-9,19-cyclolanastan-3*β*-ol (8), and 24-methylenecycloartan-3-one (9); two sterols: *β*-sitosterol (4) and stigmasterol (5); the benzofuran (2*S*,3*R*)-5-acetyl-7,3α-dihydroxy-2*β*-(1-isopropenyl)-2,3-dihydrobenzofuran (2); the 3,5-diprenyl-acetophenone (11); the fatty acids: palmitic acid (12), hexadecenoic acid, methyl ester (13), isopropyl palmitate (14), palmitamide (15), and oleamide (16); and the amino acid derivative methyl pyroglutamate (17). Compounds 1–5 and 10–14 were identified in the ethyl acetate extract. Compounds 1–5 were isolated by column chromatography, as described in Section 3.4, and identified by an analysis of their ^1^H and ^13^C-NMR data (Appendix A) and compared with the literature data [27]. Compounds 10–14 were identified by gas chromatography coupled with mass spectrometry (GC–MS) analysis (NIST 1.7a). Compounds 4 and 5 and 10 and 11 were previously reported as constituents of the aerial parts of *A. pichinchensis* [15,17], while compounds 1–3 and 12 and 13 were previously reported in a callus culture of *A. pichinchensis* [27].

The MeOH extract was fractionated by column chromatography (as described in Section 3.4), and from the chromatographic work, five groups of fractions were obtained (MSR-M-1 to MSR-M-5). The MSR-M-1 group was analyzed by GC–MS, and compounds 15–17 were identified. The MSR-M-2 and MSR-M-3 groups were purified by column chromatography. From these groups, compounds 1 and 2 were isolated and identified in mixtures of 4 and 5. The MSR-M-4 group, subjected to a purification process, allowed for obtaining three groups of fractions: MSR-M-4A (1–32; 21.2 mg), MSR-M-4B (33–42; 32.7 mg), and MSR-M -4C (43–68; 48.4 mg). The MSR-M-4A group was analyzed by GC–MS, and compounds 8 and 9 were identified. The MSR-M-4B group was subjected to silica gel column chromatography, and compound 6 was obtained as a semisolid mass. The ^1^H-NMR spectrum of (6) displayed three signals at δ 1.30 (3H, s, CH_3_-13), δ 2.44 (3H, s, CH_3_-12), and δ 3.53 (2H, s, CH_2_-14), which suggest the presence of CH_3_-, CH_3_CO, and –CH_2_OH groups in the skeleton. In the aromatic proton region, signals of an ABX system at δ 7.65 (1H, dd, *J* = 8.4, 2.2 Hz, H-7), δ 7.51 (1H, d, *J* = 2.2 Hz, H-5), and δ 6.73 (1H, d, *J* = 8.5 Hz, H-8) indicated the presence of a three-substituted aromatic ring. The olefinic AB system at δ 5.58 (1H, d, *J* = 10 Hz, H-4) and δ 6.40 (1H, d, *J* = 10 Hz, H-3) established the presence of a double bond with *cis*-configuration (Appendix A). Moreover, the ^13^C-NMR spectrum (Appendix A) showed 13 signals, including an acetoxy carbonyl carbon (δ 196.84; CO); two oxygenated carbons (δ 80.86 and 68.99; CO and CH_2_OH); five methine protons at δ 127.94, 130.97, 127.45, 116.67, and 124.32 (C-7, C-3, C-5, C-4, and C-8); four quaternary carbons at δ 157.38, 130.21, 120.64, and 80.86 (C-9, C-6, C-10, and C-2); and two methyl groups in δ 26.32 and 23.49 (C-12 and C-13). According to the data obtained for ^1^H and ^13^C-NMR, compound 6 was characterized as (-)-Artemesinol (6) [39]. Compound 7 was obtained as a semi-solid mass from the MSR-M-4C fractions group. The ^1^H-NMR spectrum of (7) displayed signals at δ 1.30 (3H, s, CH_3_-13), δ 2.44 (3H, s, CH_3_-12), δ 3.91 (1H, d, *J*=10.6 Hz, H-14a), δ 3.58 (1H, d, *J* =10.7 Hz, H-14b), and a symmetric AB system at δ 5.67 (1H, d, *J* = 10 Hz, H-3) and 6.42 (1H, d, *J* = 10 Hz, H-4), as well as hydrogens in the aromatic region at δ 7.69 (1H, dd, *J* = 8.4, 2.2 Hz, H-7), 7.56 (1H, d, *J* = 2.2 Hz, H-5), and 6.77 (1H, d, *J* = 8.5 Hz, H-8).

The ^1^H, correlated spectroscopy (COSY), DEPT, and ^13^C-NMR (Appendix A) displayed a compound similar to (-)-Artemesinol (6) but with the additional presence of a pyranose sugar moiety at δ 4.46 (1H, d, *J* = 7.8 Hz, H-1´) and δ 3.87-3.18 (6H, m, H-2´ to H-6´). The chemical shift of the proton at H-l´ (δ 4.46) and the coupling constant *J*^1´,2^´of 7. 8 Hz indicated that the glycosidic bond is β-configured. The direct connections between the protons and carbons were identified by the heteronuclear single quantum correlation (HSQC) spectrum (Appendix A). In the heteronuclear multiple bond correlation (HMBC) spectrum (Appendix A), the glycosidic linkage was located at C-14, based on the correlations between H-1´(δ_H_ 4.46, 1H, d, *J*= 7.8 Hz) and C-14 (δ_C_ 75.19), as well as the hydrogens H-14a (δ_H_ 3.91, 1H, d, *J* = 10.6 Hz) and H-14b (δ_H_ 3.58, 1H, d, *J* = 10.7 Hz), with C-2 (δ_C_ 79.95), C-13 (δ_C_ 23.79), and C-3 (δ_C_ 127.77). The optical rotation _(_α_)D_ = –8.3° (c 0.8, CH_3_OH) suggests a relative *R* configuration for C-2 of compound 7. The evidence indicates that the structure of compound 7 is 6-acetyl-(*S*)-2-methyl-2-(*R*)-glucopyranosyl chromene or Artemesinol glucopyranoside [40]. It is important to note that the 1D (^13^C, DEPT) and 2D NMR data (COSY, HSQC, and HMBC) of compound 7 have not been described in the literature. Minority compounds (8–11), identified in the *A. pichinchensis* cell culture, have been employed in cosmetic preparations [17,41,42,43].

### 2.4. Production of Bioactive Compounds During Growth Kinetics

The GC–MS analysis of the EtOAc extract obtained from the biomass of the *A. pichinchensis* cell suspension culture indicated that the main bioactive compounds (2*S*,3*R*)-5-acetyl-7,3α-dihydroxy-2*β*-(1-isopropenyl)-2,3-dihydrobenzofuran (2) and 3-epilupeol (3) are produced during all growth phases (Figure 5).

The quantification of compounds 2 and 3 was performed by analyzing the peaks at the retention time (RT) = 20.67 min (compound 2) and 38.70 min (compound 3); the molecular ion peaks were observed at m/z = 234 for 2 and 426 for 3 in GC–MS (Appendix A). The production of compound 2 started from the lag phase, and the maximum production (510.75 ± 29.10-µg g^−1^ DW) occurred on day eight of the exponential phase; then, it gradually decreased until day 22. Similar results were observed in *Capsicum chinense* Jacq. cell suspension cultures, in which the capsaicin compound reached a maximum production of 567.4-µg g^−1^ DW at 25 days [31]. On the other hand, in *Celosia cristata* cell suspension cultures, betalaine production was observed at the beginning of the exponential phase and then decreased, remaining stable during the exponential and stationary phases [30].

Regarding the 3-epilupeol compound (3), an association with the culture growth was observed, obtaining its maximum yield (410.59 ± 36.91-µg g^−1^ DW) at day 16 (Figure 5). Likewise, the production of the fatty acid amide spilanthol by cell suspension cultures of *Spilanthes acmella* Murr. presented a trend associated with its growth, reaching a maximum yield during the exponential phase and, subsequently, decreasing rapidly due to the lack of nutrients and consequent cellular death [32]. In another species, *Eurycoma longifolia*, it was reported that cell suspension cultures produce the quassinoid eurycomanone; this also occurs from the beginning of the growth kinetics, reaching a maximum amount of 1.7-mg g^−1^ DW [44].

These results surpass those reached by callus cultures, whose maximum production was identified on day 30, producing 650-µg g^−1^ DW for compound 2 and 201.10-µg g^−1^ DW of compound 3 [27]. Reducing the time of production of the compounds is a desirable characteristic in cell suspension cultures; in addition to being more homogeneous compared to callus cultures, it is possible to increase the production of bioactive compounds by adding elicitors and by scaling up reactors [45,46,47].

On the other hand, encecalin (10) (m/z 232, RT = 18.5 min) and 3,5-diprenyl-1,4-hydroxyacetophenone (11) (m/z 272, RT = 21.87 min) were detected in very low concentrations during the exponential phase (day eight) (Figure 6). Antifungal, gastroprotective, and antinociceptive effects were reported for compounds 10 and 11 [17,43]. The highlight of both compounds is that their production on in vitro cultures was reported for the first time. Based on the importance of its biological effects, cell suspension cultures of *A. pichinchensis* are a useful alternative for the production of compounds 10 and 11, which could be increased by inductors.

### 2.5. In Vitro Anti-Inflammatory Activity

The anti-inflammatory activity of the ethyl acetate extracts of the biomass of the *A. pichinchensis* suspension cell culture was assessed at different times of the growth kinetics: day 8 (D8), day 12 (D12), and day 16 (D16). Firstly, the extracts were evaluated for their effect on the viability of RAW 264.7 cells at different concentrations (5, 10, 20, 30, and 40-µg mL^−1^). No extracts exhibited a significant reduction in the viability of macrophages compared with the control group, while the positive control (etoposide) showed a significant reduction in the cellular viability at 40-µg mL^−1^ (Figure 7).

To assess the effect of the extracts from D8, D12, and D16 on nitric oxide (NO) production in lipopolysaccharide (LPS)-stimulated RAW 264.7 cells, cells were treated with the extracts at the same concentrations used in the viability assay. It was reported that LPS, an essential component of the outer membrane of Gram-negative bacteria, can induce inflammatory responses in RAW 264.7 macrophages to produce proinflammatory molecules such as NO [48,49]. The experimental results showed that the NO level was increased in LPS-stimulated RAW cells, and this effect was decreased significantly by treatment with extracts at the concentrations tested (Figure 8).

The results showed that D12 and D16 were the most active extracts that inhibited NO production at 40-µg mL^−1^, with 35.14% ± 7.55% and 34.42% ± 7.15% inhibition, respectively. On the other hand, the extract from D showed 24.69% ± 6.17% inhibition in NO production; at this point of the kinetics, compound 2 is produced in greater quantity compared to compound 3 (510.75 ± 29.10-µg g^−1^ DW and 127.85 ± 14.86-µg g^−1^ DW, respectively). In the extract from D12, the concentrations of the compounds were 434.01 ± 32.25-µg g^−1^ DW (2) and 244.89 ± 13.34-µg g^−1^ DW (3); finally, it was observed that the extract from D16 contains more of compound 3 (410.60 ± 36.91-µg g^−1^ DW), which showed a greater effect with respect to compound 2, which was identified at 335.45 ± 21.72-µg g^−1^ DW. This result corroborates the outstanding effect of compound 3: as it increases in content in the extracts, the anti-inflammatory effect also increases proportionally. However, in a pure way, both compounds have an important anti-inflammatory effect. Indomethacin (the positive control) showed an inhibition of 47.45% ± 7.41% at 30-µg mL^−1^ (Figure 8).

The results obtained in this work are of great interest, because (2*S*,3*R*)-5-acetyl-7,3α-dihydroxy-2*β*-(1-isopropenyl)-2,3-dihydrobenzofuran (2) and 3-epilupeol (3) have important pharmacological properties. Compound 2 inhibits the secretion of NO, interleukin (IL)-6, and tumor necrosis factor-alpha (TNF-α) in RAW 264.7 macrophages, as well as the activation of nuclear factor-kappa beta (NF-κB) in RAW-blue macrophages [27], while compound 3 shows antiviral [50], anti-inflammatory [51], antitubercular [52], and cytotoxic activity [53]. As to the anti-inflammatory activity, 3-epilupeol (3) shows a marked inhibition of the edema induced by**** 12-*O*-tetradecanoylphorbol-13-acetate (TPA) in mice and exhibits nitric oxide (NO) production inhibitory activity in LPS-activated macrophages [54,55,56,57]. Nitric oxide (NO) is a short-lived bioactive molecule that plays an important role in the host defense response against various pathogens, such as bacteria, viruses, fungi, and parasites, and, also, participates in pathophysiological processes such as neuronal communication, vasodilatation, and neurotoxicity [58,59].

The importance of the anti-inflammatory effects of 3-epilupeol (3) lies in the fact that the excessive production of NO causes tissue damage, extensive systemic vasodilatation, and hypotension [54]. In addition, NO is involved in inflammatory disorders, including bowel diseases, rheumatoid arthritis, chronic hepatitis, pulmonary fibrosis, and colon cancer [56,58,59,60].

## 3. Materials and Methods

### 3.1. General Procedures

Compounds 1–7 were isolated by column chromatography (silica gel 70–230 mesh; Merck, Darmstadt, Germany), and the purity of the compounds was checked by gas chromatography coupled with mass spectrometry (GC–MS); detection was performed at ultraviolet (UV) light (366 and 254 nm) and after spraying with Ce(SO_4_)_2_ 2 (NH_4_)2SO_4_ 2H_2_O (Sigma-Aldrich, Inc., Toluca, State of México, Mexico), followed by heating. The ^1^H- and ^13^C-; DEPT; and 2D nuclear magnetic resonance (NMR) (correlated spectroscopy (COSY), heteronuclear single quantum correlation (HSQC), and heteronuclear multiple bond correlation (HMBC)) experiments were recorded on a Varian Unity Inova 200 MHz and a Bruker AVANCE IIIHD 500 at 500 MHz, using CDCl_3_ and CD_3_OD with tetramethylsilane (TMS) as the internal standard. Optical rotations were measured on a 241 digital polarimeter at 25 °C (Perkin Elmer, Waltham, MA, USA), equipped with a sodium lamp (589 nm) and microcell. Indomethacin (indo), dimethyl sulfoxide (DMSO), etoposide, lipopolysaccharide (LPS) from *Escherichia coli* serotype 055: B5, sodium nitrite (NaNO_2_), phosphoric acid (H_3_PO_4_), N-(1-naphtyl) ethylenediamine dihydrochloride, and sulfanilamide were purchased from Sigma-Aldrich. DMEM/F12 (Dulbecco’s modified Eagle’s medium/Nutrient Mixture F-12) and fetal bovine serum (FBS) were from GIBCO (Waltham, MA, USA). (3-(4,5-dimethyl-2-yl)-5-(3-carboxymethoxyphenyl)-2-(4-sulfophenyl)-2H-tetrazolium, inner salt; MTS) was from Promega Co. (Fitchburg, WI, USA). Murine macrophage cell line RAW 264.7 (Tib-71™) was from ATCC® (Washington, DC, USA). Compounds 8–15 were identified by GC–MS. Compounds 2 and 3 were quantitated by GC–MS using a calibration curve of authentic samples following the method described for the *A. pichinchensis* callus culture [27].

### 3.2. Plant Material

Plants and seeds of *Ageratina pichinchensis* (Kunth) were collected in Tepoztlán Morelos, Mexico in March 2018 (19°00’43.88” N, 99.05’38.66” W), identified by Gabriel Flores Franco, and deposited at the HUMO Herbarium of the Universidad Autónoma del Estado de Morelos (UAEM), with the voucher number 33913.

### 3.3. Establishment of Cell Suspension Cultures

Friable callus cultures of *A. pichinchensis* were previously established by our working group [27]. Calluses were subcultured in the same Murashige and Skoog (MS) semisolid culture medium containing 3% sucrose, 1.0-mg L^−1^ α-naphthaleneacetic acid (NAA), and 0.1-mg L^−1^ 6-furfurylaminopurine (KIN). The 20-day-old calluses were used to establish the cell suspension cultures. Fresh calluses (5 g) were transferred to 250-mL Erlenmeyer flasks containing 50 mL of MS liquid culture medium using the same conditions and plant growth regulator as in the callus cultures. Cell suspension cultures were placed on an orbital shaker at 110 rpm and incubated at 25 ± 2 °C under a photoperiod of 16-h with white fluorescent light (50-µmol m^–2^ s^−1^). When an increase of biomass was shown in the flasks, cells were harvested and screened with 200-µm nylon mesh filters (Whatman No. 1) to obtain a homogeneous cell culture. To increase the biomass, the cells were subcultured every 15 days for six months using an inoculum size of 10% (v/v) in 500-mL Erlenmeyer flasks with 100 mL of liquid culture medium.

#### 3.3.1. Growth Kinetics

The growth kinetics of the cell suspension culture was carried out in 250-mL flasks without modifying the composition of the culture medium. Each flask containing 50 mL of medium MS was inoculated with 2 g of fresh cells and incubated in the same conditions mentioned above. Three flasks were harvested every two days, and the culture was allowed to grow for 22 days. Harvested cells were washed with distilled water, filtered with a cellulose filter (Whatman No. 1), and dried in an oven at 55 °C for 24 h. Then, dry-weight biomass (DW) data were recorded to perform the culture growth curve. The specific cell growth rate (µ) was calculated by plotting the natural logarithm of the cell growth data versus time. The doubling time (Dt) was computed from the µ exponential data.

#### 3.3.2. Cell Viability

The cell viability of the cell suspension culture was measured by Evan’s blue day exclusion method [61]. A sample of 1 mL of cell suspension was taken from each flask and incubated into 0.25% Evan’s blue stain for 5 min, and at least 700 cells were counted. Viable cells were considered those that were not stained. All the experiments were repeated three times, with three replicates each.

#### 3.3.3. Sugar Quantification and pH Measurement

During each sampling of growth kinetics, 5-mL aliquots from the residual culture medium of each biomass sample were taken; their pH was measured with a potentiometer (Science Med SM-25CW) and total sugar content by the phenol–sulfuric method [62]. A calibration curve was performed using sucrose as a standard at concentrations of 0.1 to 1.0-µg mL^−1^. A sample aliquot (2 mL) of the sample was mixed with 2 mL of phenol reagent at 5% in digester tubes and placed in a rack submerged in a cold-water bath. Then, 5 mL of H_2_SO_4_ concentrated was added to the mixture and allowed to stand for 15 min and analyzed in a spectrophotometer at 490 nm against a blank.

### 3.4. Extraction and Isolation of Compounds from Cell Suspension Cultures

Biomass harvested on day 20 was dried in an oven at 40 °C (12.30 g) and extracted with 100 mL of ethyl acetate by sonication (30 min); the extraction process was performed in triplicate. The excess of solvent was eliminated in a rotatory evaporator under reduced pressure, and a brown residue (1.3 g) was obtained. A second extraction was carried out with methanol (3 × 100 mL); the solvent was removed by distillation, giving rise to a resinous residue (2.4 g).

The ethyl acetate extract (1.2 g) was fractioned in an open chromatographic column previously packed with silica gel (36 g, 70–230 mesh, Merck) and eluted with a *n*-hexane, ethyl acetate gradient system (100:00, 95:05, 90:10, 85:15, 80:20, 75:25, and 00:00) v/v). Fractions of 20 mL were collected to obtain 64 fractions that were monitored by thin-layer chromatography (TLC) (ALUGRAM® SIL G/UV254 silica gel plates). Fractions that showed TLC similarity were grouped, obtaining six groups: MSR-EA-1 (1–9; 241.4 mg), MSR-EA-2 (10–18; 138.4 mg), MSR-EA-3 (19–32; 122.3 mg), MSR-EA-4 (33–42; 147.2 mg), MSR-EA-5 (43–55; 216.5 mg), and MSR-EA-6 (56–64; 270.6 mg). The MSR-EA-1 fraction showed a single spot in TLC, and its GC–MS analysis indicated that this fraction consisted of a mixture of palmitic acid (12), hexadecenoic acid methyl ester (13), and isopropyl palmitate (14). The MSR-EA-2 fraction (120 mg), obtained in an *n*-hexane/ethyl acetate system (95:05), was purified using column chromatography, silica gel (4 g), and an *n*-hexane/ethyl acetate gradient system ( 98:02 → 86: 14). In total, 28 fractions of 3 mL each were obtained; in fractions 16–22, obtained in a *n*-hexane/ethyl acetate system (92:08), a single spot was observed in TLC, and these were combined to obtain 18.3 mg (0.1%) of a colorless oil. Analysis of ^1^H and ^13^C-NMR data allowed us to identify desmethoxyencecalin (1). The MSR-EA-3 fraction (120 mg) was purified by column chromatography, silica gel, and a system of gradients with *n*-hexane/ethyl acetate (96:04 → 80:20). From this purification process, 53 fractions were obtained. The mixture of stigmasterol (4) and *β*-sitosterol (5) (41.8 mg; 0.3%) was obtained as a white amorphous solid in the fractions 42–48, eluted with *n*-hexane/ethyl acetate (87:13). An aliquot (115 mg) of the MSR-EA-4 group was purified by column chromatography. Silica gel was used as the stationary phase, and a *n*-hexane/ethyl acetate gradient system (90:10 → 80:20) was used as the mobile phase. Forty-eight fractions were obtained, and the purification process was monitored by TLC, which allowed us to create five groups of fractions (MSR-EA-4A to MSR-EA-4E). In the MSR-EA-4B (10.3 mg) group obtained in the 86:14 system (*n*-hexane/ethyl acetate), a mixture of sterols (4) and (5) was identified; from the group MSR-EA-4C (37.6 mg) obtained in 84:16 (*n*-hexane/ethyl acetate), a crystalline solid was obtained. It was analyzed by ^1^H and ^13^C-NMR, and its data allowed us to identify it as 3-epilupeol (3). The MSR-EA-4D group (8 mg) obtained in an 82:18 system (*n*-hexane/ethyl acetate) was analyzed by GC-MS, and the compounds 24-methylene-9,19-cyclolanastan-3*β*-ol (8) and 24-methylenecycloartan-3-one (9) were identified. Column chromatographic purification of the MSR-EA-5 fraction (216.5 mg), using a *n*-hexane–EtOAc gradient system (95:05 → 70: 30), provided 64 fractions. These were combined according to their chemical profiles observed in TLC in seven groups of fractions: MSR-EA-5A (1–20; 34.6 mg), MSR-EA-5B (21–25; 23.4 mg), MSR-EA-5C (26–33; 19.2 mg), MSR-EA-5D (34–42; 16.9 mg), MSR-EA-5E (43–50; 18.1 mg), MSR-EA-5F (51–58; 28.7 mg), and MSR-EA-5G (59–64; 44.2 mg). Fractions 51–58 eluted with 76:24 *n*-hexane-EtOAc contained a viscous liquid, and a single spot was observed on TLC. This was analyzed by ^1^H and ^13^C-NMR, and the analysis of their data led to the identification of the compound as (2*S*,3*R*)-5-acetyl-7,3α-dihydroxy-2*β*-(1-isopropenyl)-2,3-dihydrobenzofuran (2).

#### 3.4.1. Desmethoxyencecalin (1)

Compound (1) was isolated as a colorless oil. ^1^H-NMR (200 MHz, CDCl_3_) δ*:* 7.69 (1H, dd, *J* = 8.5, 2.2, Hz, H-7), 7.56 (1H, d, *J* = 2.2 Hz, H-5), 6.76 (1H, d, *J* = 8.5 Hz, H-8), 6.45 (1H, d, *J* = 9.9 Hz, H-4), 5.58 (1H, d, *J* = 10 Hz, H-3), 2.47 (3H, s, CH_3_ of Ac), 1.40 (2 CH_3_, s, C- 11 and C-12). ^13^C-NMR (50 MHz, CDCl_3_) 196.54 (CO of Ac), 157.29 (C-10), 131.24 (C-6), 130.72 (C-7), 127.54 (C-5), 127.46 (C-3), 124.56 (C-4), 120.67 (C-9), 116.24 (C-8), 77.67 (C-2), 26.51 (CH_3_, C-14), 23.36 (2CH_3_, C-11, and C-12)). These data match those in the literature [27]. Spectra of ^1^H and ^13^C-NMR are in Appendix A.

#### 3.4.2. (2S,3R)-5-Acetyl-7,3α-dihydroxy-2β-(1-isopropenyl)-2,3-dihydrobenzofuran (2)

Compound (2) was isolated as a yellowish oil. ^1^H-NMR (200 MHz, CDCl_3_) δ: 7.53 (1H, d, *J* = 1.4 Hz, H-6), 7.44 (1H, d, *J* = 1.4 Hz, H-4), 5.13 (1H, d*, J* = 3.8 Hz, H-3), 5.03 (1H, d*, J* = 1.2 Hz, H-12a), 4.96 (1H, d, *J* = 3.8 Hz, H-2), 4.89 (1H, d, *J* = 1.2 Hz, H-12b), 2.46 (3H, s, H-14), 1.67 (3H, s, H-11). ^13^C-NMR (50 MHz, CDCl_3_): 197.36 (C-13), 151.89 (C-8), 140.92 (C-7), 140.67 (C-10), 132.37 (C-5), 129.33 (C-9), 118.85 (C-4), 117.91 (C-6), 113.54 (C-12), 95.78 (C-2), 76.88 (C-3), 26.69 (C-14), and 17.56 (C-11). These data match those in the literature [27]. Spectra of ^1^H and ^13^C NMR are in Appendix A.

#### 3.4.3. 3-Epilupeol (3)

Compound (3) was isolated as a crystalline solid. ^1^H-NMR (200 MHz, CDCl_3_) δ: 4.58 (1H, d, *J* = 2.2 Hz, 29a); 4.56 (1H, d, *J* = 1.2 Hz, 29b); 3.26 (1H, t, *J* = 2.8 Hz, H-3); 2.19 (1H, td, *J* = 11, 5.9 Hz, H-19); 1.88 (2H, c, H-21); 1.66 (2H, m, H-2); 1.65 (3H, s, H-30); 1.63 (2H, t, H-22); 1.52 (2H, m, H-16); 1.40 (1H, m, H-18); 1.42 (1H, m, H-5); 1.37 (2H, m, H-1, H-15); 1.36 (6H, m, H-7, H-11, H-12,); 1.34 (2H, m, H-6); 1.27 (1H, m, H-13); 1.19 (1H, t, H-9); 1.10 (6H, s, CH_3_-23 and 24); 0.98 (3H, s, CH_3_-25); 0.91 (3H, s, CH_3_-26); 0.82 (3H, s, CH_3_-28); 0.84 (3H, s, CH_3_-27). ^13^C-NMR (50 MHz, CDCl_3_) δ: 151.23 (C-20); 109.51 (C-29); 76.47 (C-3); 50.45 (C-9); 49.25 (C-5); 48.53 (C-18); 48.25 (C-19); 43.23 (C-17); 43.08 (C-14); 41.18 (C-8); 40.26 (C-22); 38.18 (C-13); 37.69 (C-4); 37.51 (C-10); 35.81 (C-16); 34.38 (C-7); 33.28 (C-1); 30.06 (C-21); 28.48 (C-23); 27.60 (C-15); 25.63 (C-2); 25.33 (C-12); 22.38 (C-24); 21.12 (C-11); 19.48 (C-30); 18.53 (C-6); 18.24 (C-28); 16.17 (C-25); 16.12 (C-26); and 14.92 (C-27). These data match those in the literature [27]. Spectra of ^1^H and ^13^C NMR are in Appendix A.

#### 3.4.4. Stigmasterol (4) and *β*-Sitosterol (5)

Compounds (4) and (5) were isolated as white amorphous solids. δ: ^1^H-NMR (200 MHz, CDCl_3_) *δ*: 5.13 (1H, t, *J* = 7.8 Hz, H-6); 5.07 (2H, m, H-22, H-23); 3.52 (2H, m, H-3); 1.62 to 1.95 (4H, m, H-7); 1.49 to 1.55 (2H, m, H-8); 0.85 to 0.93 ( 2H, m, H-9); 1.47 to 1.55 (4H, m, H-11); 1.19 to 1.23 and 1.91 to 2.03 (4H, m, H-12); 1.01 to 1.17 (2H, m, H-14); 1.48 to 1.60 (2H, m, H-25); 0.85 to 1.85 ( 22H, m, H-15 to H-24 and H-28); 0.92 (CH_3_, d, *J* = 7.2 Hz, H-27); 0.94 (CH_3_, d, *J* = 6.8 Hz, H-26); 0.98 (CH_3_, d, *J* = 6.4 Hz, H-21); 0.87 (CH_3_, s, H-18); 1.09 (CH_3_, s, H-19). ^13^C-NMR (50 MHz, CDCl_3_) *δ*: 37.5 (CH_2_, C-1); 31.9 (CH_2_, C-2); 72.02 (CH, C-3); 40.45 (CH_2_, C-4); 140.76 (C, C-5); 121.72 (CH, C-6); 32.18 (CH_2_, C-7); 32.34 (CH, C-8); 50.46 (CH, C-9); 36.38 (C, C-10); 21.43 (CH_2_, C-11); 39.86 (CH_2_, C-12); 42.74 (C, C-13) 56.77 and 57.03 (CH, C-14); 24.32 and 24.61 (CH_2_, C-15); 28.25 and 28.51 (CH_2_, C-16); 57.08 (CH, C-17); 12.10 (CH_3_, C-18); 19.68 (CH_3_, C-19); 36.28 and 40.14 (CH, C-20); 18.21 and 19.10 (CH_3_, C-21); 34.21 and 138.76 (CH_2_ and CH, C-22); 26.38 and 129.50 (CH_2_ and CH, C-23); 46.07 and 51.37 (CH, C24); 29.4 (CH, C-25); 19.31 and 21 04 (CH_3_, C-26); 19.30 and 19.23 (CH_3_, C-27); 23.02 and 23.54 (CH_2_, C-28); and 12.26 (CH_3_, C-29). These data match those in the literature, and authentic samples are available in our laboratory [27].

MeOH extract (2.4 g) was fractionated in an open chromatographic column previously packed with silica gel (72 g, 70–230 mesh; Merck) and eluted with a CH_2_Cl_2_:MeOH gradient system (100:00, 95:05, 90:10, 85:15, 80:20, 75:25, and 00:100, v/v). Fractions of 50 mL were collected to obtain 118 fractions and monitored by TLC. Fractions that showed similarity in the TLC analysis were grouped into five groups: MSR-M-1 (1–60; 548.3 mg), MSR-M-2 (61–83; 218.6 mg), MSR-M-3 (84–86; 314.8 mg), MSR-M-4 (87–95; 112.6 mg), and MSR-M-5 (96–118; 878.4 mg). The MSR-M-1 group was analyzed by GC–MS, and the compounds palmitamide (15), oleamide (16), and methyl pyroglutamate (17) were identified. By successive chromatography of the fraction MSR-M-2, using silica gel as the stationary phase and a system of gradients (95:05 → 88:12) with *n*-hexane/ethyl acetate, compound 1 (8 mg) was identified. Fractions 84–86 (MSR-M-3; 314.8 mg) were obtained in a gradient system of increasing polarity with CH_2_Cl_2_: MeOH (90:10 → 85: 15); this fraction was purified by column chromatography, silica gel, and an *n*-hexane/ethyl acetate elution system (90:10 → 80:20). Of the fractions obtained in the *n*-hexane/ethyl acetate system (86:14), in the mixture of sterols, we identified 4 and 5 (46.8 mg), and in the fractions obtained from the (82:18) system, compound 2 (6 mg) was obtained. The MSR-M-4 fraction (112.6 mg) was purified by column chromatography and silica gel and eluted with a CH_2_Cl_2_:MeOH (98:02 → 70:30) gradient system; 68 fractions were obtained. Fractions that showed TLC similarity were grouped into three groups: MSR-M-4A (1–32; 21.2 mg), MSR-M-4B (33–42; 32.7 mg), and MSR-M-4C (43–68; 48.4 mg). The MSR-M-4A group was analyzed by GC–MS, and compounds 24-metilene-9,19-cyclolanastan-3*β*-ol (8) and 24-methylenecycloartan-3-one (9) were identified. The MSR-M-4B fraction (32.2 mg) obtained in a CH_2_Cl_2_/MeOH system (88:12 → 82:18) was purified by column chromatography, silica gel, and an isocratic system with CH_2_Cl_2_/MeOH (84:16), obtaining a semisolid (14 mg). Analysis by ^1^H and ^13^C-NMR allowed the compound to be identified as (-)-Artemesinol (6). The MSR-M-4C fraction (48.4 mg) obtained in a CH_2_Cl_2_/MeOH system (80:20 → 74:26) was purified by column chromatography, silica gel, and an isocratic system with CH_2_Cl_2_/MeOH (75:25), obtaining a semisolid (34 mg). Analysis of NMR in 1D (^1^H and ^13^C) and 2D (COSY, HSQC, and HMBC) allowed the compound to be identified as (-)-Artemesinol glucoside (7).

#### 3.4.5. (-)-Artemesinol (6)

Compound (6) was isolated as a semi-solid mass. ^1^H NMR (200 MHz, CDCl_3_:CD_3_OD) δ 7.65 (1H, dd, *J* = 8.4, 2.2 Hz, H-7), 7.51 (1H, d, *J* = 2.2 Hz, H-5), 6.73 (1H, d, *J* = 8.5 Hz, H-8), 6.40 (1H, d, *J* = 10 Hz, H-4), 5.58 (1H, d, *J* = 10 Hz, H-3), 3.53 (2H, s, CH_2_-14), 2.44 (3H, s, CH_3_-12), and 1.3 (3H, s, CH_3_-C-O). ^13^C NMR (50 MHz, CDCl_3_:CD_3_OD) δ 196.84 (C-11), 157.38 (C-9), 130.21 (C-6), 127.94 (C-7), 130.97 (C-3), 127.45 (C-5), 116.67 (C-4), 120.64 (C-10), 124.32 (C-8), 80.86 (C-2), 68.99 (C-14), 26.32 (C-12), and 23.49 (C-13). Spectra of ^1^H and ^13^C NMR are in Appendix A. Stereochemistry at C-2 was not determined, but a relative (*R*) configuration was suggested, compared to the optical rotation (α)_D_ = –4.5 ° (c 0.89, CHCl_3_) reported for this compound [39].

#### 3.4.6. (±)-Artemesinol Glucoside (7)

Compound (**7**) was isolated as a semisolid mass. ^1^H NMR (500 MHz, CDCl_3_:CD_3_OD) δ 7.69 (1H, dd, *J* = 8.4, 2.2 Hz, H-7), 7.56 (1H, d, *J* = 2.2 Hz, H-5), 6.77 (1H, d, *J* = 8.5 Hz, H-8), 6.42 (1H, d, *J* = 10 Hz, H-4), 5.67 (1H, d, *J* = 10 Hz, H-3), 4.46 (1H, d, *J* = 7.8 Hz, H-1´), 3.91 (1H, d, *J* = 10.6 Hz, H-14a), 3.87-3.18 (6H, m, H-2´ to H-6´), 3.58 (1H, d, *J* = 10.7 Hz, H-14b), 2.48 (3H, s, CH_3_-12), and 1.39 (3H, s, CH_3_-13). The ^1^H NMR data were compared with those reported [41]. ^13^C NMR (126 MHz, CDCl_3_:CD_3_OD) δ 197.44 (C-11), 157.31 (C-9), 130.71 (C-6), 130.66 (C-7), 127.77 (C-3), 127.30 (C-5), 123.71 (C-4), 120.69 (C-10), 116.12 (C-8), 103.57 (C-1´), 79.55 (C-2), 76.18 (C-5´), 75.90 (C-3´), 75.19 (C-14), 73.51 (C-2´), 69.85 (C-4´), 62.19 (C-6´), 26.39 (C-12), and 23.79 (C-13). Spectra of 1H, 13C, COSY, DEPT, HSQC, and HMBC are in Appendix A.

Compounds 8 and 9 and 12–17 were identified by comparing the GC relative retention times and MS fragmentation pattern of a single compound with those from the NIST 1.7a mass spectral library. GC–MS chromatograms of compounds 8 and 9 and 12–17 are in Appendix A. Compounds 10 and 11 were identified in the suspension cells by comparing their GC relative retention times and MS fragmentation patterns. The standards used in the analysis were as follows: encecalin (10) and 3,5-diprenyl-4-hydroxyacetophenone (11), previously isolated from *A. pichinchensis*. GC–MS chromatograms are in Appendix A.

### 3.5. Quantification of Compounds 2 and 3 by GC-MS

Biomass collected at 0, 2, 4, 6, 8, 10, 12, 14, 16, 18, 20, and 22 days were dried in an oven at 55 °C for 24 h. Subsequently, each sample was extracted by sonication with EtOAc (25 mL × 3) and MeOH (25 mL × 3) and concentrated in a rotatory evaporator. Maximum production of compounds 2 and 3 was identified in ethyl acetate extracts on days 8 (D8) and 16 (D16), respectively. For the quantitative analysis, a standard curve of compounds 2 and 3 was prepared in triplicate and analyzed by GC–MS. Concentrations of 2.2, 1.1, 0.55, 0.275, 0.1375, and 0.06875-mg mL^−1^ were used for compound 2 and of 0.350, 0.175, 0.0875, 0.04375, and 0.02187-mg mL^−1^ for compound 3. Each standard solution was analyzed in triplicate to calculate the peak area ratio (y) and relative concentration (x); these data were used to construct a linear calibration curve, which showed acceptable linearity with correlation coefficients r^2^ = 0.9926 and 0.9997, respectively (Appendix A). The quantification of compounds 2 and 3 in the extracts was expressed as µg g^−1^ biomass dry weight (µg g^−1^ DW).

### 3.6. In Vitro Anti-Inflammatory Activity

#### 3.6.1. Cell Culture

RAW 264.7 cells were maintained in DMEM/F12 medium supplemented with 10% heat-inactivated FBS, without antibiotics. Cells were cultured at 37 °C in a humidified atmosphere containing 5% CO_2_ and subcultured by scraping and seeding them in 25-cm^2^ flasks of 96-well plates [52,53].

#### 3.6.2. Assay for Cell Viability

Cells (1 × 10^4^ cells/well in 100 µL of medium) were seeded in a 96-well plate and incubated for 24 h. Then, the cells were incubated for 22 h in the presence of extracts at various concentrations (5–40 µg mL^−1^) or vehicle (DMSO, 0.21%, v/v) or etoposide (40-µg mL^−1^), which served as a positive control, and cells without treatment were considered a negative control. Cell viability was determined by adding MTS solution to each well and incubating the cells for another 2 h. The optical density was measured at 490 nm on a microplate reader.

#### 3.6.3. Treatment with LPS

Cells (2 × 10^4^ cells/well in 200 µL of medium) were plated and incubated for 24 h in 96-well plates. After that, the cells were incubated for 1 h in the presence of extracts at noncytotoxic concentrations (5 to 40-µg mL^−1^), a vehicle (DMSO, 0.21%, v/v), or indomethacin (30-µg mL^−1^), which served as a positive control; cells without treatment were considered a negative control. Then, the cells were incubated at 37 °C for 20 h with LPS at 4-µg mL^−1^ (for wells with extracts, vehicle, indomethacin, and 100% stimulus control) as a proinflammatory stimulus and without LPS (negative control). Finally, cell-free supernatants were collected and used for NO quantification.

#### 3.6.4. Determination of NO Concentration

Nitrite, the stable end product of NO, was used as an indicator of NO production in the cell-free supernatants and was measured according to the Griess reaction. Briefly, 50 µL of each supernatant were mixed with 100 µL of Griess reagent (50 µL of 1.0% sulfanilamide and 50 µL of 0.1% N-(1-naphtyl)ethylenediamine dihydrochloride in 2.5% phosphoric acid solution) in a new 96-well plate and incubated for 10 min at room temperature. The optical density at 540 nm (OD_540_) was measured with a microplate reader, and the nitrite concentrations in the samples were calculated by comparison with the OD_540_ of a standard curve of NaNO_2_ in a fresh culture medium [27,57,60].

### 3.7. Statistical Analysis

The results shown were obtained from at least three independent experiments and are presented as the means ± standard deviation. Statistical analysis was performed by one-way analysis of variance (ANOVA), followed by Dunnett’s multiple comparisons test. For suspension cell cultures, the values of each experiment are means, and the bars represent the standard error of triplicate determinations. All statistical analyses were performed using GraphPad Prism® version 8.0 software; *p*-values < 0.5 were considered to indicate statistical significance. Microsoft® Excel® for Office 364 MSO (16.0.11425.20242) 32-bit software was used for analysis.

## 4. Conclusions

The establishment of a cell suspension culture of *A. pichinchensis* is reported for the first time. This cell suspension culture retained the ability to produce the anti-inflammatory compounds 2,3-dihydrobenzofuran (2) and 3-epilupeol (3) identified previously in a callus culture from this species. Moreover, the yield of both compounds was improved, and the production time was reduced by almost half compared with callus cultures. Phytochemical analysis of the cell cultures led to the identification of 17 bioactive components, of which compounds 2–5 and 8–11 were previously described to have anti-inflammatory, antimicrobial, antifungal, and gastroprotective properties. Furthermore, the in vitro anti-inflammatory efficacy of the cell culture extracts and the identification for the first time of (-)-Artemesinol (6), (-)-Artemesinol glucoside (7), encecalin (10), and 3,5-diprenyl-acetophenone (11) in cell cultures of *A.*
*pichinchensis* offer a biotechnological tool for bioreactor scale-up to produce anti-inflammatory compounds in a sustainable way.

## Figures and Tables

**Figure 1 plants-09-01398-f001:**
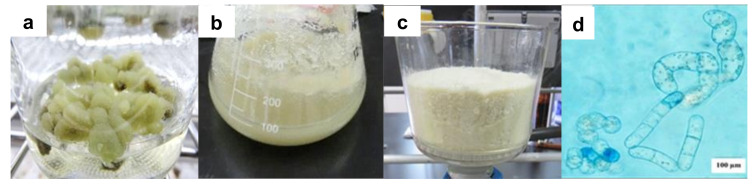
Cell culture from *Ageratina pichinchensis* leaves. (**a**) Friable callus at 20 days of culture; (**b**) cell suspension culture with abundant biomass at 16 days of culture; (**c**) filtered biomass; (**d**) microscopic image of cell suspension culture (100×).

**Figure 2 plants-09-01398-f002:**
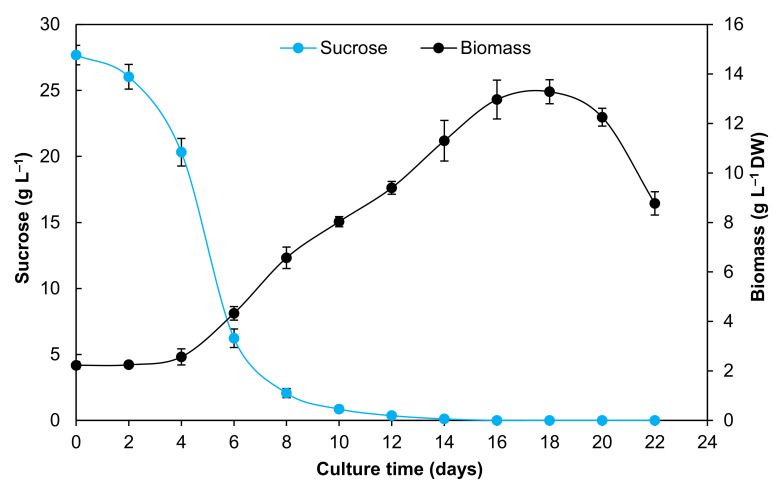
Growth kinetics and consumption of sucrose of the *A. pichinchensis* cell suspension culture during growth for 22 days in Murashige and Skoog (MS) medium with 1.0-mg L^−1^ α-naphthaleneacetic acid (NAA) and 0.1-mg L^−1^ 6-furfurylaminopurine (KIN).

**Figure 3 plants-09-01398-f003:**
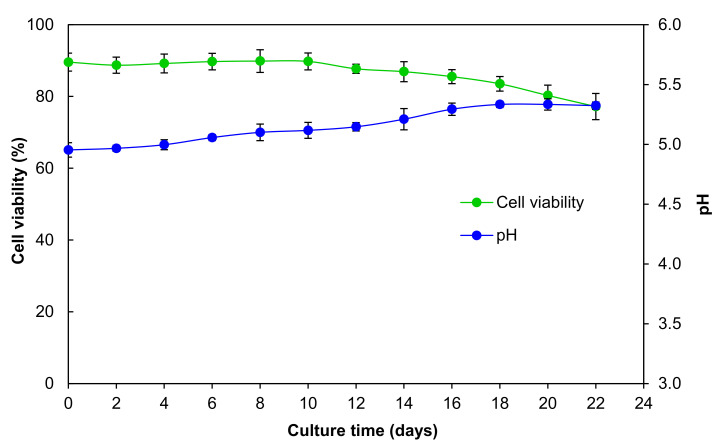
Viability of the cell suspension culture and pH of the culture medium over 22 days of culture growth of *A. pichinchensis*.

**Figure 4 plants-09-01398-f004:**
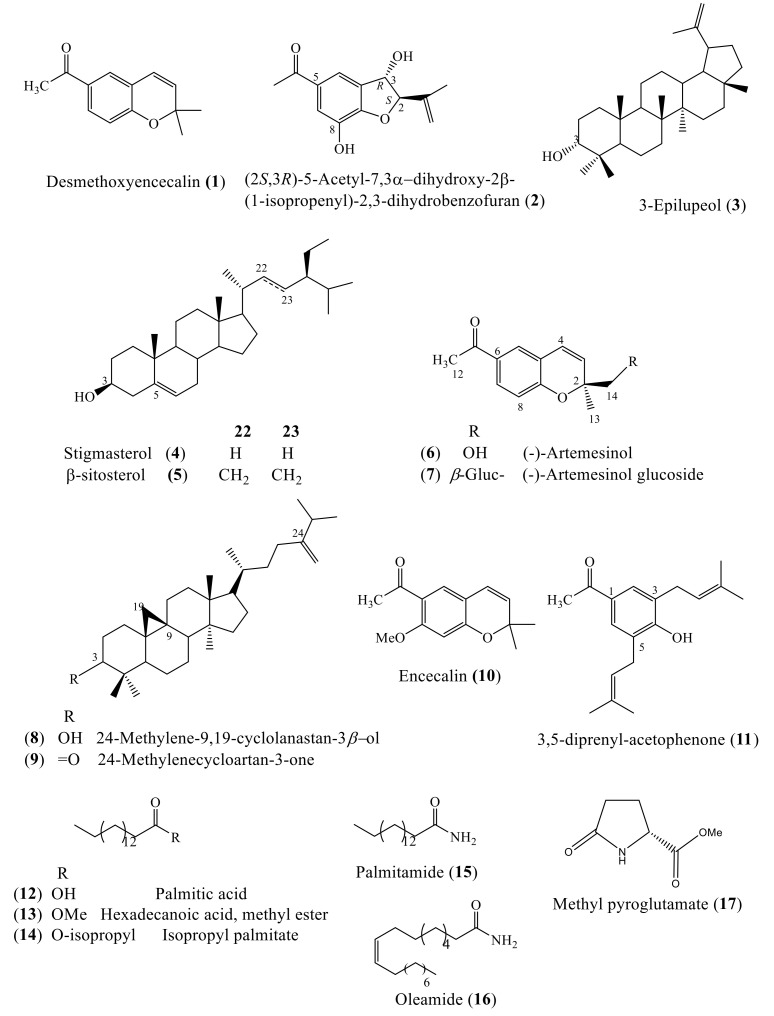
Chemical structures of compounds identified in the cell suspension culture of *A. pichinchensis.*

**Figure 5 plants-09-01398-f005:**
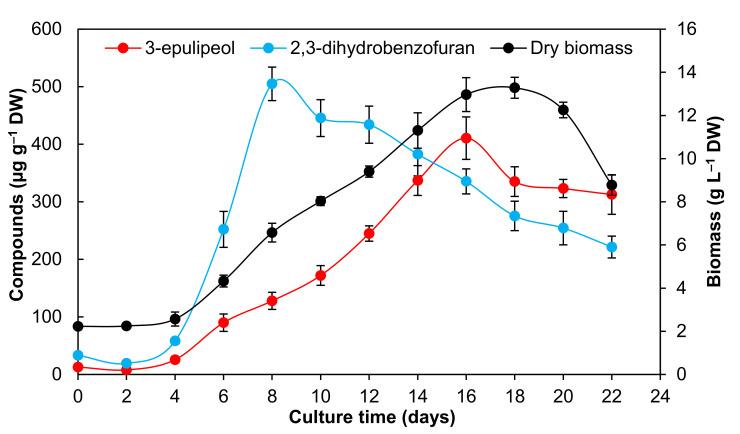
Relationship between the production of 2,3-dihydrobenzofuran and 3-epilupeol and the culture growth of *A. pichinchensis* over 22 days. DW: dry weight.

**Figure 6 plants-09-01398-f006:**
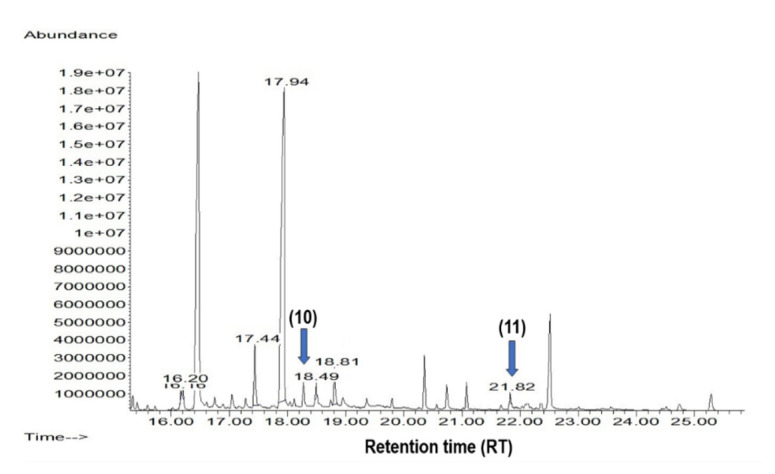
Gas chromatography coupled with mass spectrometry (GC-MS) chromatogram of the ethyl acetate extract from day 8 of the suspension cell cultures, showing the encecalin (10) and 3,5-diprenyl-4-hydroxyacetophenone (11) compounds.

**Figure 7 plants-09-01398-f007:**
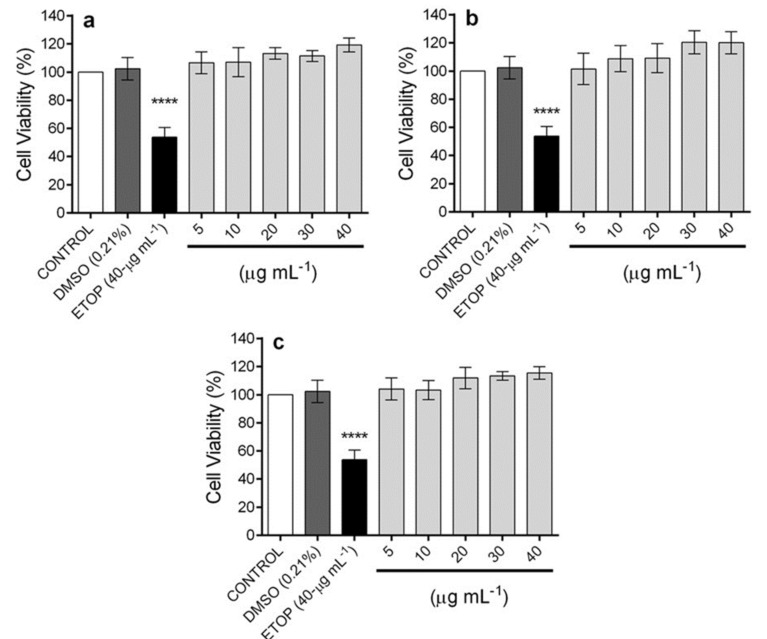
Effect of ethyl acetate extracts from *A. pinchichensis* cell suspension cultures on the viability of RAW 264.7 cells. (**a**) extract from day 8 (D8); (**b**) extract from day 12 (D12); (**c**) extract from day 16 (D16). The values are expressed as the mean ± SD of three independent experiments (*n* = 3). Significance was determined using ANOVA, followed by Dunnett’s multiple comparisons test (*****p* < 0.0001 dimethyl sulfoxide (DMSO), ETOP (etoposide), and extracts compared with the control group).

**Figure 8 plants-09-01398-f008:**
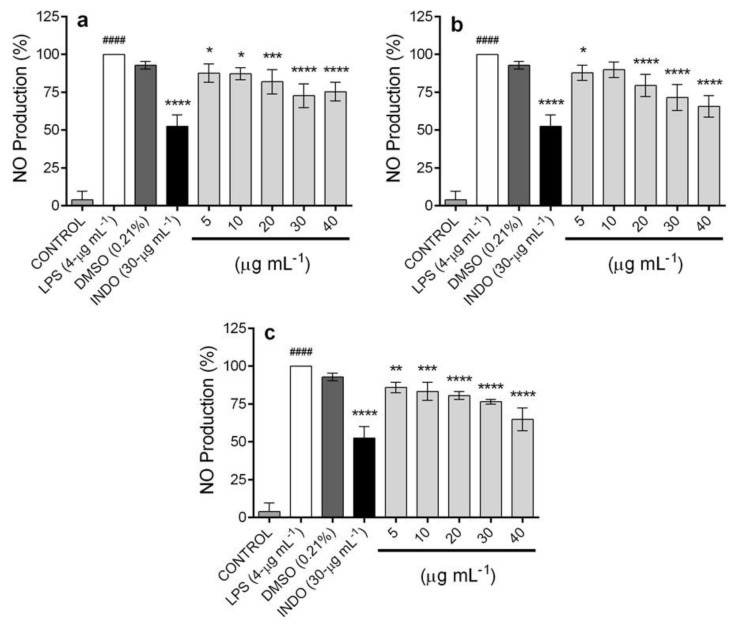
Effect of ethyl acetate extracts from *A. pinchichensis* cell suspension cultures on the nitric oxide (NO) production in lipopolysaccharide (LPS)-stimulated RAW 264.7 macrophages. (**a**) extract from day 8 (D8); (**b**) extract from day 12 (D12); (**c**) extract from day 16 (D16). The values are expressed as the mean ± SD of three independent experiments (*n* = 3). Significance was determined using ANOVA followed by Dunnett’s multiple comparisons test (####*p* < 0.0001 LPS compared with the control group; **p* < 0.05 and ****p* < 0.0001 DMSO, INDO (indomethacin), and extracts compared with the LPS group).

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
