# Peer review of "Establishment of a Cell Suspension Culture of Ageratina pichinchensis (Kunth) for the Improved Production of Anti-Inflammatory Compounds"

_plants, 2020, doi:10.3390/plants9101398_

Round 1

Reviewer 1 Report

The manuscript entitled "Establishment of a Cell Suspension Culture of Ageratina pichinchensis for the Improved Production of Anti-inflammatory Compounds" raises the interesting problem of trying to obtain valuable biologically active compounds from plant cells. Those with anti-inflammatory properties are particularly important, because a wide range of diseases is accompanied by a strong inflammatory process, which in turn may lead to permanent changes in the diseased tissue.
The authors of the manuscript did not adhere to the basic requirements of manuscript preparation, which resulted in its poor quality.
The discussion and results should be separated which would allow for a more readable form of the manuscript. Although the idea of trying to obtain selected anti-inflammatory compounds from plant suspension cell cultures is interesting, this form of presentation (including English language and style) does not allow for a recommendation for publication.
I have marked other minor comments in the PDF file

Author Response

Response to Reviewer 1

Reviewer: The manuscript entitled "Establishment of a Cell Suspension Culture of Ageratina pichinchensis for the Improved Production of Anti-inflammatory Compounds" raises the interesting problem of trying to obtain valuable biologically active compounds from plant cells. Those with anti-inflammatory properties are particularly important, because a wide range of diseases is accompanied by a strong inflammatory process, which in turn may lead to permanent changes in the diseased tissue.

The authors of the manuscript did not adhere to the basic requirements of manuscript preparation, which resulted in its poor quality.

Authors: The entire manuscript was reviewed and prepared according to Instructions for Authors.

Reviewer: The discussion and results should be separated which would allow for a more readable form of the manuscript.

Authors: All authors reviewed the manuscript and we agreed that it is more appropriate to combine the Results and Discussion. According to the Instructions for Authors, both sections can be combined. In fact, there are several Articles published in this way, for example: Plants 2020, 9, 1111; doi:10.3390/plants9091111; Plants 2020, 9, 1130; doi:10.3390/plants9091130; Plants 2020, 9, 950. Moreover, we reviewed and improved the entire manuscript for better readability.

Reviewer: Although the idea of trying to obtain selected anti-inflammatory compounds from plant suspension cell cultures is interesting, this form of presentation (including English language and style) does not allow for a recommendation for publication.

Authors: We reviewed and improved the entire manuscript; moreover, the style and language were reviewed and edited by a Native-speaking English. Certificate is attached.

Reviewer: Line 17. PDF file, “due to its healing properties”

Authors: The abstract was slightly rewritten and shortened to 200 words.

Reviewer: Line 18. PDF file, “plant name's author is missing”

Authors: Plant name’s author was added. This was also added within the main Title, Abstract and in the Plant Material section. (Lines 1, 16 and 300).

Reviewer: Line 24. PDF file. unfinished sentence

Authors: The paragraph was rewritten. Lines 22-25.

Reviewer: Line 35. PDF file, “ant”.

Authors: “ant” (line 35) was substituted by “and”. Line 35.

Reviewer: Lines 46-53 “a very long and therefore not very clear sentence”.

Authors: The paragraph was rewritten. Lines 46-52. Moreover, the entire manuscript was checked.

Reviewer: Line 64. Results and Discussion should be separated.

Authors: This observation has already been answered above.

Reviewer: Line 332, “H2SO4”.

Authors: This was corrected (line 337). Moreover, all superscripts and subscripts were revised.

Reviewer: Line 450. “104”

Authors: The superscripts “104” were corrected. Line 501.

Reviewer: Lines 507-5690. all references should be formatted as per journal requirements

Authors: All References were reviewed and corrected according to the Instructions for Authors.

Reviewer 2 Report

  1. Publish with minor revision

    In general, this manuscript established a culture method for cell suspension of A. pichinchensis, with which the production of anti-inflammatory compounds has been improved. In this study, 17 known compounds were identified, and their structures were assigned with NMR and GC-MS. The production of 4 compounds has been reported for the first time by suspension cell cultures of A. pichinchensis. In general, the manuscript descripts the idea clearly and provided comprehensive analysis on the identified compounds. In the section of “Results and discussion”, the broader application of the technique can be addressed more. Also please discuss whether the method is applicable to larger batch production. Together with following correction list, the manuscript is suitable for publishing with minor revision.

    Minor revision

    1. Page 1 line 25, missing period.
    2. Please provide the full name of abbreviations such as GC, MS when they first used in the manuscript.
    3. Page 3, the figure 2 is deformed especially the text labeled on the figure. Please use another original figure.
    4. Page 4, line 102, change to “At the beginning…”.
    5. Page 4, line 114, change to “ in the pH values at the exponential phase”.
    6. Page 8, figure 6 is with low resolution and the figure is deformed. Please replace with another figure with higher resolution.

Author Response

Response to Reviewer 2

Reviewer: In general, this manuscript established a culture method for cell suspension of A. pichinchensis, with which the production of anti-inflammatory compounds has been improved. In this study, 17 known compounds were identified, and their structures were assigned with NMR and GC-MS. The production of 4 compounds has been reported for the first time by suspension cell cultures of A. pichinchensis. In general, the manuscript descripts the idea clearly and provided comprehensive analysis on the identified compounds. In the section of “Results and discussion”, the broader application of the technique can be addressed more. Also please discuss whether the method is applicable to larger batch production. Together with following correction list, the manuscript is suitable for publishing with minor revision.

Authors: The Results and Discussion section was revised and improved.

Reviewer: 1. Page 1 line 25, missing period.

Authors: The paragraph was corrected. Lines 22-25.

Reviewer: 2. Please provide the full name of abbreviations such as GC, MS when they first used in the manuscript.

Authors: This was done. All abbreviations were described in Materials and Methods and where it was necessary. These were no longer mentioned in the rest of the manuscript to avoid repeating.

Reviewer: 3. Page 3, the figure 2 is deformed especially the text labeled on the figure. Please use another original figure.

Authors: This was done. The figure was replaced by the original figure. Page 3.

Reviewer: 4. Page 4, line 102, changed by “At the beginning…”.

Authors: The paragraph was rewritten (lines 99-101).

Reviewer: 5. Page 4, line 114, change by “in the pH values at the exponential phase”.

Authors: The entire paragraph was rewritten (lines 113-119).

Reviewer: 6. Page 8, figure 6 is with low resolution and the figure is deformed. Please replace with another figure with higher resolution.

Authors: This was done. The Figure 6 was replaced by a better resolution figure. Page 8.

Reviewer 3 Report

The manuscript by Sánchez-Ramos et al. reports on the production of  bioactive phytochemicals in suspension cells of Ageratina pichinchensis. In particular, a well growing suspension culture was obtained from callus cultures and the phytochemical profile of cell extracts was depicted by identifying 17 known compounds; two of them were quantified during the cell growth cycle. The anti-inflammatory activity of cell extracts was also assessed using animal cells.  The subject is of interest and the results could add some new knowledge to the area. However, the manuscript needs to be improved.

- The main concern regards the English requiring a deep revision throughout the whole manuscript. Some sentences seem incomplete, thus the meaning is hardly clear (L. 24-25; 115-116; 118-119; 206-207;...). L. 101; 105: “Which” is not properly used.  Please correct and clarify. Other sentences are too long and difficult to read through (L. 46-53; 208-214; 243-252).

- Fig. 3: changes of pH or viability do not seem so pronounced; are they statistically significant?

-Fig. 4 should show results of phytochemical analysis but is just descriptive of the 17 identified compounds with no data on the identification. A table including  other information could be more suitable.

- Fig. 5: are “dry biomass” data the same as “biomass” in fig. 2? If so, better to specify and use here a dashed line.

- L.200: (2) should be (3)

- Regarding  the in vitro anti-inflammatory activity, please add some information and a reference for the use of RAW 264.7 cells.

- Conclusions should be better focused. “Compound 2 was obtained on day 8... and compound 3... on day 16” is misunderstanding since they were also obtained earlier or later (Fig.5).

Author Response

Response to Reviewer 3

The manuscript by Sánchez-Ramos et al. reports on the production of bioactive phytochemicals in suspension cells of Ageratina pichinchensis. In particular, a well growing suspension culture was obtained from callus cultures and the phytochemical profile of cell extracts was depicted by identifying 17 known compounds; two of them were quantified during the cell growth cycle. The anti-inflammatory activity of cell extracts was also assessed using animal cells. The subject is of interest and the results could add some new knowledge to the area. However, the manuscript needs to be improved.

Reviewer: - The main concern regards the English requiring a deep revision throughout the whole manuscript.

Authors: A Native-speaking English reviewed and edited the entire manuscript. Certificate is attached.

Reviewer: Some sentences seem incomplete, thus the meaning is hardly clear (L. 24-25; 115-116; 118-119; 206-207;...). L. 101; 105: “Which” is not properly used. Please correct and clarify. Other sentences are too long and difficult to read through (L. 46-53; 208-214; 243-252).

Authors: We corrected the marked sentences. The current corrected lines are: 24-26, 46-52, 99-101, 106-107, 113-119, 122-123, 206-215, 252-260. In addition, we reviewed the entire manuscript for better understanding.

Reviewer: - Fig. 3: changes of pH or viability do not seem so pronounced; are they statistically significant?

Authors: It is low significant; however, the pH of the culture medium can affect cell viability, in addition, maintaining good viability indicates an adequate cell culture establishment. More details are mentioned in the manuscript (lines 105-127)

Reviewer: -Fig. 4 should show results of phytochemical analysis but is just descriptive of the 17 identified compounds with no data on the identification. A table including other information could be more suitable.

Authors: The identification data was added as figures in the supplementary material. The phytochemical work of the ethyl acetate extract from cell culture is described in the Materials and Methods section (Lines 339-380). Of this work, compounds 1-5, 8-9 and 12-14 were identified. Compounds 1-5 were identified by data of 1H, 13C NMR and compared to reported. Data 1H, 13C NMR are attached on lines 381-421 and the mass spectra are in Supplementary Figure S1‒S6. Compounds 8-9, 12-14 and 15-17 were identified by comparing the relative retention times (GC) and fragmentation pattern (MS) with the library NIST 1.7a (Supplementary Figures S15, S16 and S17).

            Compounds 10 and 11 were identified by comparing their relative retention times (GC) and fragmentation pattern (MS) with the standards: encecalin (10) and 3,5-diprenyl-4-hydroxyacetophenone (11) previously isolated from Ageratina pichinchensis. The GC-MS chromatograms were added in Supplemental Figure S18.

Lines 145 to 181. “J” was changed by “J” (italic).

Line 159. “S1” was changed by “Figure S7”

Line 160. “S2” was changed by “Figure S8”

Line 171. “S3, S4, S5 and S6” was changed by “Figures S9-S12”, respectively.

Lines 173-174.   “ -1´ ” was changed by “ H-1´ ” and “ J1´,2´ “ was change by “ J1´,2´ “ (italic J and superscript numbers).

Line 175. “S7” was changed by “Figure S13”.

Line 176. “S8” was changed by “Figure S14”.

Lines 177-179. “δH, δC” was change by “δH, δC (H and C as subscript).

Line 179. “[α]20D” was changed by “[α]D (as subscript)

Lines 363. “An Aliquot of 115 mg of column chromatography” was changed by “An aliquot (115 mg) of MSR-EA-4 group was purified by column chromatography”.

Line 381-387. The subheading “3.4.1. (-)-Artemesinol (6)” was replaced by “3.4.1.  Desmethoxyencecalin (1)” and Data 1H, 13C NMR was added.

Line 388. The subheading “3.4.2. (±)-Artemesinol Glucoside (7)” was replaced by “3.4.2. (2S,3R)-5-Acetyl-7,3α-dihydroxy-2β-(1-isopropenyl)-2,3-dihydrobenzofuran (2)” and Data 1H, 13C NMR was added.

Line 395-406. The subheading “3.4.3. 3-Epilupeol (3)” and Data 1H, 13C NMR were added.

Lines 408-422. The subheading “3.4.4. Stigmasterol (4) and β-Sitosterol (5)” and Data 1H, 13C NMR   were added.

Line 349. Subheading numbering “3.4.1.” was changed by “3.4.5.”

Line 455. “S1 and S2” was changed by “Figures S7,S8”, respectively.

Line 456. “[α]D“ was changed by “ [α]D “.    

Line 458. Subheading numbering “3.4.2.” was replaced by “3.4.6.”.

Line 467. Figures “S3 to S8” were changed by “S9-S14”.

Lines 468-474. The identification of compounds 8-9, 10, 11 and 12-17 was added.

Line 485. Supplementary Figures “S9,S10” was substituted by “Figures S19,S20”, respectively.

Lines 537-548. Supplementary Figures “S1-S8” was changed by “S7-S14”; “S9-S10” was substituted by “S19-S20”, data “S1” by “S6” and “S15” by “S18”.

Reviewer: - Fig. 5: are “dry biomass” data the same as “biomass” in fig. 2? If so, better to specify and use here a dashed line.

Authors: This was done.

Reviewer: - L.200: (2) should be (3)

Authors: This was corrected. Line 202.

Reviewer: - Regarding the in vitro anti-inflammatory activity, please add some information and a reference for the use of RAW 264.7 cells.

Authors: Quotes and References were added. Lines 242, 492 and 685-690.

Reviewer: - Conclusions should be better focused. “Compound 2 was obtained on day 8... and compound 3... on day 16” is misunderstanding since they were also obtained earlier or later (Fig.5).

Authors: The Conclusions section were rewritten and the focus was improved.

Round 2

Reviewer 1 Report

I accept the manuscript as it stands.